# Training-free Detection of AI-generated Images via High-Frequency Influence

## Abstract

Dramatic advances in the quality of the latent diffusion models (LDMs) also led to the malicious use of AI-generated images. While current AI-generated image detection methods assume the availability of real/AI-generated images for training, this is practically limited given the vast expressibility of LDMs. This motivates the training-free detection setup where no related data are available in advance. In this paper, we propose that the level of aliasing detected in reconstructed images produced by the autoencoder of LDMs can serve as a criterion for distinguishing between real and AI-generated images. Specifically, we propose a novel detection score function, termed high-frequency influence (HFI), which quantifies the impact of the spatial filtering-based high-frequency components of the input image on the perceptual reconstruction distance. HFI is training-free, efficient, and consistently outperforms other training-free methods in detecting challenging images generated by various generative models. We also show that HFI can successfully detect the images generated from the specified LDM. HFI outperforms the best baseline method while achieving magnitudes of speedup.

## 1 Introduction

With the rapid advancement of generative AI, we are now able to generate photorealistic images in a desired context within seconds. The crucial factor in this achievement is the emergence of foundational Latent Diffusion Models (LDMs) (MidJourney, 2022; Rombach et al., 2022). LDMs represent the integration of recent breakthroughs of vision models, including the powerful generation quality of diffusion models (Song & Ermon, 2019; Song et al., 2021), the joint text-image representation learned by vision-language pre-trained models (Radford et al., 2021), and the improved inference speed through compression in the latent space (Kingma & Welling, 2014). However, with the progression of such models, we should also consider the potential negative impacts such generated images may have on society. For example, LDMs can produce fake images that could cause societal confusion (Yim, 2024) or infringe upon intellectual property rights (Brittain, 2024). Moreover, applying AI-generated images on training generative models may deteriorate model performance (Alemohammad et al., 2024). Therefore, AI-generated image detection or deepfake detection methods, which aim to distinguish AI-generated images from real images, are gaining significant attention.

However, most AI-generated image detection methods adhere to traditional settings that are disconnected from real-world challenges. As shown in Figure 1, these methods follow a training-based setup. Namely, they are designed to train on data sampled from a given real image distribution with similar AI-generated data, and they are tested on the data sampled from the same real image distribution against other AI-generated data. While such a setup was effective when generative models were typically designed to fit a single specific dataset, it faces practical limitations when we aim to detect LDM-generated data in general, which are trained on billions of images (Schuhmann et al., 2022). Moreover, such LDMs can generate hallucinated images that may not have been encountered in reality (The Guardian, 2024), which induces extra effort to acquire matching real images.

We explore a training-free AI-generated image detection framework that assumes no access to training images as an alternative to traditional setups. A few methods in this domain (Ricker et al., 2024; He et al., 2024) aim to design a universal score function that distinguishes AI-generated images from real ones by leveraging representations from large-scale pre-trained models. For example, Ricker et al. (2024) and He et al. (2024) leverage the representation of the autoencoder of LDM and

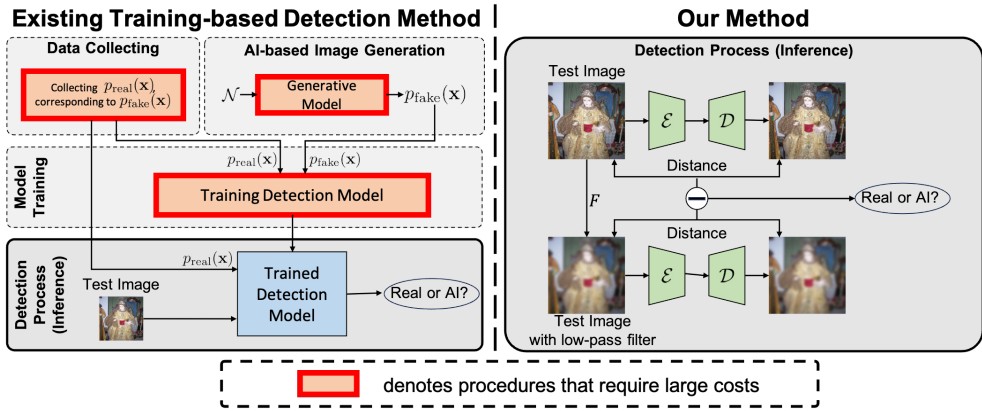

Figure 1: **Problem setup of HFI. (Left)** Setup of training-based AI-generated image detection methods. Such methods train and test on the same real data distribution. Furthermore, the framework can be costly when detecting images produced by large-scale text-to-image generative models. **(Right)** Pipeline of our proposed HFI. HFI operates only on the test time and can be computed efficiently via the autoencoder of the LDM.

Dinov2 (Oquab et al., 2024), respectively. While these methods are directly deployable in real-world scenarios without further training, none of the methods have shown their efficacy on challenging benchmarks (*e.g.*, GenImage (Zhu et al., 2023)) that contain LDM-generated images, or images generated by other text-to-image generative models (*e.g.*, GLIDE (Nichol et al., 2022)).

**Contributions.** Due to the limitations of traditional downsampling-upsampling architectures, which are affected by aliasing in pixel prediction tasks (Agnihotri et al., 2024; Chen et al., 2024b), we examine whether autoencoders of the LDM can effectively compress and recover high-frequency information from a novel data. Based on the observation that autoencoders often misrepresent high-frequency details in real data, we propose our detection score function, high-frequency influence (HFI). HFI measures the impact of spatial filtering-based high-frequency components on the distance between the input and its reconstruction via the autoencoder. As shown in Fig 1, HFI can be implemented at test time with various choices of distance functions and low-pass filters available.

We demonstrate the efficacy of HFI through extensive experiments in challenging AI-generated image detection benchmarks in various domains (*e.g.*, natural images (Zhu et al., 2023), face (Chen et al., 2024c)). HFI outperforms existing training-free methods consistently. Moreover, HFI further reduces the gap against the competitive training-based method DRCT (Chen et al., 2024a) *i.e.*, $5.7\% \rightarrow 1.7\%$. We also apply HFI on the task of tracing model-generated AI images given a specific LDM model. We show that HFI outperforms the state-of-the-art method (Wang et al., 2024) while being 57x faster.

In brief, our contributions are summarized as follows.

- We propose HFI, a novel score function that distinguishes AI-generated images from real images without any training (Section 3).
- HFI outperforms existing baselines on challenging benchmarks (Section 4.2). We conduct extensive ablation studies for potential improvement of HFI (Section 4.3).
- We also apply HFI into the distinct setup of detecting model-generated images from the specified model (Section 5). HFI improves over the baseline (Wang et al., 2024) with significant speedup.

## 2 PRELIMINARIES

### 2.1 PROBLEM SETUP

Given the real data distribution $p_{\text{real}}(\mathbf{x})$, the manipulator can utilize the generative model $G_i$ to produce deepfake distribution $p_{\text{fake},G_i}(\mathbf{x})$ that is similar to $p_{\text{real}}(\mathbf{x})$. The goal of the AI-generated image detection is to design a score function $U(\mathbf{x})$ that determines whether $\mathbf{x}$ is from $p_{\text{real}}(\mathbf{x})$ (*i.e.*, $U(\mathbf{x}) > \tau$) or not (*i.e.*, $U(\mathbf{x}) \leq \tau$). We denote "A *vs* B" as an AI-generated image detection task

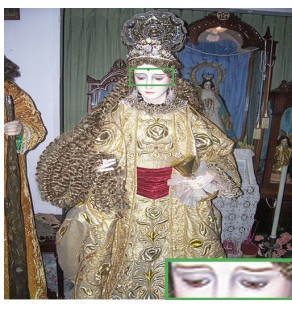 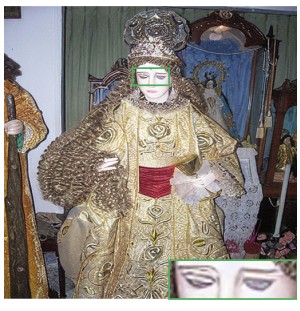 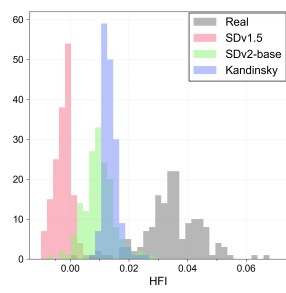

(a) Original image  (b) Reconstructed image  (c) Histogram of HFI

Figure 2: **Motivation of HFI.** **(a)** Sampled data from the ImageNet (Deng et al., 2009) dataset. **(b)** Reconstruction through the autoencoder of the Stable Diffusion (Rombach et al., 2022) v1.1 model. We can observe obvious distortions in the high-frequency details. **(c)** Histogram of HFI experimented in toy dataset. We utllize the autoencoder of SDv1.4 for computing HFI.

where A is the real dataset and B is the applied generative model to mimic dataset A. For clarity, we note that the term "real image" in this paper does not include samples in the training dataset of B.

This paper studies training-free AI-generated image detection where universal metric design is required without prior knowledge. Namely, we do not have access to $p_{real}(\mathbf{x})$ or $p_{fake,G_i}(\mathbf{x})$. Given the diverse expressibility of concurrent text-to-image generation models (*e.g.*, Stable Diffusion (SD) (Rombach et al., 2022)), our setup matches the goal of real-world AI-generated image detection.

## 2.2 LATENT DIFFUSION MODELS

LDMs (Rombach et al., 2022) efficiently generate high-dimensional images by modeling the diffusion process on the latent space $\mathcal{Z} \subset \mathbb{R}^{C' \times H' \times W'}$ instead of the data space $\mathcal{X} \subset \mathbb{R}^{C \times H \times W}$ with $C = 3$. Hence, LDMs first pre-train the encoder $\mathcal{E} : \mathcal{X} \to \mathcal{Z}$ and the decoder $\mathcal{D} : \mathcal{Z} \to \mathcal{X}$. This paper will refer to LDM as a comprehensive framework for autoencoder-based text-to-image generative models.

While the autoencoder of the LDM is defined as VAEs (Kingma & Welling, 2014), weight on the KL-regularization loss during training is negligible, resulting in negligible variance in the latent space (Rombach et al., 2022). Hence, we view the encoder and the decoder as the deterministic mapping between $\mathcal{X}$ and $\mathcal{Z}$ throughout the paper. We set $AE(\mathbf{x}) := \mathcal{D} \circ \mathcal{E}(\mathbf{x})$ for the rest of the paper.

## 2.3 ATTRIBUTION OF LDM-GENERATED IMAGES.

A recent line of works (Wang et al., 2023b; 2024) aims to detect data generated from the given LDM model $\mathcal{M}$ as a means of alternative to explicit watermarking (Fernandez et al., 2023; Wen et al., 2023). In specific, for a given LDM model $\mathcal{M}_1$, existing methods design an uncertainty score function that distinguishes the belonging data generated from $\mathcal{M}_1$ and the non-belonging data generated from the other generative model $\mathcal{M}_2$. We denote such task as "$\mathcal{M}_1$ vs $\mathcal{M}_2$" model attribution task.

## 3 METHOD

Our motivation is grounded in the phenomenon of aliasing (Gonzalez & Woods, 2006), which occurs when an original signal contains high-frequency components exceeding the subsampling rate, leading to distortions in the downsampled signal and subsequently affecting its upsampled reconstruction. Recent studies (Agnihotri et al., 2024; Chen et al., 2024b) have also explored aliasing within pixel-wise prediction tasks, where images are typically downsampled and upsampled through convolutional layers, analogous to the functioning of the autoencoder in LDMs. In this work, we propose that the extent of aliasing present in reconstructed images can serve as a reliable metric for distinguishing between real and AI-generated images.

We show our motivation with the reconstruction of the real image. Figure 2a and 2b show the sample of ImageNet (Deng et al., 2009) and its reconstruction through the SDv1.1 autoencoder, respectively.

The encoder $\mathcal{E}$ fails to compress high-frequency components of the real image and causes deviation. For example, we can observe obvious distortions in the high-frequency details in the reconstructed images (*e.g.*, eye, ring, pattern in cloths).

Motivated by the observations, we propose to measure the influence of the input high-frequency components on the discrepancy between the input data and its reconstruction as a detection score function. We propose our HFI as follows

$$\text{HFI}_{d,\mathcal{F},\text{AE},\nu}(\mathbf{x}) := \left\langle \frac{\partial d(\mathbf{x}, \text{AE}(\nu, \mathbf{x}))}{\partial \mathbf{x}} \ , \ \mathbf{x} - \mathcal{F}(\mathbf{x}) \right\rangle \tag{1}$$

where $d : \mathcal{X} \times \mathcal{X} \to \mathbb{R}_0^+$ is the reconstruction distance function and $\mathcal{F} : \mathcal{X} \to \mathcal{X}$ are low-pass filters, where the autoencoders (*i.e.*, $\text{AE}(\mathbf{x})$) are trained on original dataset $\nu$:

$$\text{AE}(\nu, \cdot) = \underset{\mathcal{D},\mathcal{E}}{\arg\min} \, \mathbb{E}_{\mathbf{y} \sim \nu} \left[ \left|\left| \mathcal{D} \circ \mathcal{E}(\mathbf{y}) - \mathbf{y} \right|\right|^2 \right] \tag{2}$$

The newly suggested score function is designed to capture infinitesimal distortions of reconstructed images sampled from the **test dataset** through the lens of autoencoders (*i.e.*, $\text{AE}(\mathbf{x})$) pre-trained on the **train dataset** $\nu$ (*e.g.*, LAION-aesthetics (Schuhmann et al., 2022)). By taking a directional derivative in the direction of $\mathbf{x} - \mathcal{F}(\mathbf{x})$, the score function amplifies the difference (*i.e.*, distribution shift) between the train and test dataset in high-frequency information. As conventional encoders $\mathcal{E}$ inevitably produce compression errors in the high-frequency domain, we argue that the geometric direction to $\mathbf{x} - \mathcal{F}(\mathbf{x})$, that estimates the difference of high-frequency information, is an optimal choice to reveal the failure behavior of autoencoders.

On the other hand, on text-to-image generative models that are trained on large-scale data distribution similar to $\nu$, we expect the HFI score to be lower on their generated data. Furthermore, our design of Eq 1 is also motivated by architectural similarities of generative models. Generative models employ upsampling layers composed of a series of convolution kernels (Rombach et al., 2022; Dhariwal & Nichol, 2021). We anticipate that our spatial term, $\mathbf{x} - \mathcal{F}(\mathbf{x})$, will capture these local correlations, resulting in lower HFI scores.

Unfortunately, the numerical estimation of the gradient in Eq 1 might be challenging since both components of the distance function are dependent on the input. Hence, we estimate the numerical approximation of HFI by taking the 1st-order Taylor series expansion.

$$\left\langle \frac{\partial d(\mathbf{x}, \text{AE}(\nu, \mathbf{x}))}{\partial \mathbf{x}} \ , \ \mathbf{x} - \mathcal{F}(\mathbf{x}) \right\rangle \approx d(\mathbf{x}, \text{AE}(\mathbf{x})) - d(\mathcal{F}(\mathbf{x}), \text{AE}(\mathcal{F}(\mathbf{x}))) \tag{3}$$

We expect HFI can effectively discriminate the real data (*i.e.*, $\text{HFI}(\mathbf{x}) > \tau$) or generated data (*i.e.*, $\text{HFI}(\mathbf{x}) \leq \tau$). We verify our hypothesis on the toy dataset. To be specific, we collect 162 ImageNet (Deng et al., 2009) data of the class "Jaguar" and generate the image via the text prompt "A photo of a jaguar". We use the SDv1.5, SDv2-base (Rombach et al., 2022), and Kandinsky (Razzhigaev et al., 2023) models for the generation, and the autoencoder of SDv1.4 model for the evaluating HFI, respectively. We use the LPIPS (Zhang et al., 2018) for the distance function $d$. Since the optimal filter choice is unknown, a Gaussian kernel with kernel size $k = 3$ and standard deviation $\sigma = 0.8$ is selected for the experiment as a de facto choice. The choice of $k = 3$ is informed by the architecture of the autoencoders in LDMs, which utilize $3 \times 3$ kernels.

We report the results in Figure 2c. HFI successfully distinguishes real image distribution from the AI-generated data. HFI also successfully distinguishes SDv1.5-generated data from other AI-generated data (*e.g.*, SDv2-base, Kandinsky), given that SDv1.4 and SDv1.5 share the same autoencoder.

When the autoencoders from distinct models, $\text{AE}_1, ..., \text{AE}_n$, are available, we follow the practice of (Ricker et al., 2024) and finalize the ensemble version of HFI as follows.

$$\text{HFI}_{d,\mathcal{F}}(\mathbf{x}) = \min_i \text{HFI}_{d,\mathcal{F},\text{AE}_i,\nu_i}(\mathbf{x}) \tag{4}$$

Notably, the ensemble approach aligns with real-world scenarios, where the optimal autoencoder for a given dataset is typically unknown in advance.

Table 1: Mean AI-generated image detection performance (AUROC/AUPR) and average rank of HFI and AEROBLADE (Ricker et al., 2024) in the GenImage (Zhu et al., 2023) dataset under the cross-autoencoder setup. **Bold** denotes the best method.

| | AE: SDv1.4 | | AE: SDv2-base | | AE: Kandinsky | | AE: MiniSD | |
|---|---|---|---|---|---|---|---|---|
| Method | Mean | Avg Rank | Mean | Avg Rank | Mean | Avg Rank | Mean | Avg Rank |
| AEROBLADE$_{LPIPS}$ | 0.886/0.880 | 4.00/3.88 | 0.829/0.830 | 3.63/3.63 | 0.777/0.779 | 3.63/3.63 | 0.660/0.650 | 3.38/3.25 |
| AEROBLADE$_{LPIPS_2}$ | 0.920/0.918 | 2.88/2.88 | 0.857/0.886 | 3.25/3.00 | 0.809/0.818 | 3.38/3.38 | 0.667/0.663 | 3.00/3.13 |
| HFI$_{LPIPS}$ (ours) | 0.941/0.934 | **1.38/1.38** | 0.899/0.899 | 1.63/1.75 | **0.895/0.900** | **1.38/1.38** | 0.717/0.700 | 2.13/2.00 |
| HFI$_{LPIPS_2}$ (ours) | **0.951/0.955** | 1.50/1.63 | **0.905/0.917** | **1.50/1.50** | 0.878/0.893 | 1.63/1.63 | **0.752/0.736** | **1.75/1.63** |

Table 2: Mean AI-generated image detection performance (AUROC/AUPR) and average rank (lower is better) of HFI and AEROBLADE (Ricker et al., 2024) in the DiffusionFace (Chen et al., 2024c) dataset under the cross-autoencoder setup. **Bold** denotes the best method.

| | AE: SDv1.4 | | AE: SDv2-base | | AE: Kandinsky | | AE: MiniSD | |
|---|---|---|---|---|---|---|---|---|
| Method | Mean | Avg Rank | Mean | Avg Rank | Mean | Avg Rank | Mean | Avg Rank |
| AEROBLADE$_{LPIPS}$ | 0.750/0.730 | 2.75/2.75 | 0.709/0.688 | 3.00/2.75 | 0.651/0.631 | 2.75/2.75 | 0.601/0.583 | 3.25/3.25 |
| AEROBLADE$_{LPIPS_2}$ | 0.729/0.713 | 3.50/3.25 | 0.705/0.684 | 3.75/3.50 | 0.648/0.627 | 3.75/3.50 | 0.614/0.592 | 2.75/2.75 |
| HFI$_{LPIPS}$ (ours) | **0.772/0.770** | **1.75/1.75** | 0.732/**0.729** | **1.50/1.50** | **0.727/0.723** | **1.25/1.50** | **0.693/0.677** | 2.00/2.00 |
| HFI$_{LPIPS_2}$ (ours) | 0.753/0.743 | 2.00/2.25 | **0.736**/0.724 | 1.75/2.00 | 0.690/0.676 | 2.25/2.25 | 0.692/**0.678** | 2.00/2.00 |

## 4 EXPERIMENT

In this section, we evaluate the efficacy of our proposed HFI in detecting images generated by text-to-image generative models. First, we introduce our experiment protocols and the datasets (Section 4.1). We then present our main results (Section 4.2) followed by extensive ablation studies (Section 4.3).

### 4.1 EXPERIMENTAL SETUPS

**Evaluation protocols.** Our main experiments are divided into two scenarios. We first report the performance between the autoencoder-based methods in the cross-autoencoder setup. To be specific, we compare HFI against the autoencoder-based baseline, AEROBLADE (Ricker et al., 2024), under various autoencoder choices. We use the autoencoder of the Stable diffusion (SD)v1.4/v2-base (Rombach et al., 2022), Kandinsky2.1 (Razzhigaev et al., 2023), and MiniSD-diffusers (Labs, 2022) throughout the experiment. Then, we compute the ensemble performance of HFI and AEROBLADE and report their performance against the other AI-generated image detection methods.

**Metric.** We mainly report the area of the region under the Precision-Recall curve (AUPR) and the area of the region under the ROC curve (AUROC). We additionally report the average rank across the dataset in the cross-autoencoder setup.

**Datasets.** We test the efficacy of HFI applied in various data domains. For the natural image domain, we experiment on the GenImage (Zhu et al., 2023) benchmark where the real image is from the ImageNet (Deng et al., 2009). GenImage constitutes data generated by 8 generative models that are not limited to LDMs: ADM (Dhariwal & Nichol, 2021), BigGAN (Brock et al., 2018), GLIDE (Nichol et al., 2022), Midjourney (MidJourney, 2022), SD v1.4/1.5 (Rombach et al., 2022), VQDM (Gu et al., 2022b), and Wukong (Gu et al., 2022a).

We also evaluate HFI on the DiffusionFace (Chen et al., 2024c) benchmark where the real image is from the Multi-Modal-CelebA-HQ (Xia et al., 2021) dataset. The benchmark contains 4 LDM-generated categories: images generated from the noise via SDv1.5/2.1 models (SD1.5 T2I, SD2.1 T2I) and images generated from image-to-image translation starting from real images via SDv1.5/2.1 models (SDv1.5 I2I, SDv2.1 I2I). Notably, SDv1.5 I2I and SDv2.1 I2I correspond to SDEdit (Meng et al., 2021) where styles of the real image are still present in the generated image.

**Baselines.** First, we compare HFI against AEROBLADE (Ricker et al., 2024) in the cross-autoencoder setup. Since original AEROBLADE uses the variant of LPIPS (Zhang et al., 2018) that is based on the 2nd layer of VGG (Simonyan & Zisserman, 2015) network, we denote it as AEROBLADE$_{LPIPS_2}$ and also experiment our HFI on the distance, denoted as HFI$_{LPIPS_2}$. We also report the performance of AEROBLADE based on the full LPIPS distance, denoted as AEROBLADE$_{LPIPS}$.

Table 3: AI-generated image detection performance (AUPR) of HFI and baselines in the GenImage (Zhu et al., 2023) dataset. **Bold** and underline denotes the best and second best methods.

| Method | ADM | BigGAN | GLIDE | Midj | SD1.4 | SD1.5 | VQDM | Wukong | Mean |
|---|---|---|---|---|---|---|---|---|---|
| *Training-based Detection Methods* | | | | | | | | | |
| DRCT/UnivFD | 0.892 | 0.924 | 0.964 | 0.974 | 0.997 | 0.995 | 0.966 | 0.994 | 0.963 |
| NPR | 0.733 | 0.920 | 0.924 | 0.822 | 0.842 | 0.841 | 0.766 | 0.814 | 0.833 |
| *Training-free Detection Methods* | | | | | | | | | |
| RIGID | 0.790 | 0.976 | 0.964 | 0.797 | 0.698 | 0.699 | 0.860 | 0.708 | 0.812 |
| AEROBLADE$_{LPIPS}$ | 0.732 | 0.906 | 0.973 | 0.833 | 0.966 | 0.968 | 0.577 | 0.972 | 0.866 |
| AEROBLADE$_{LPIPS_2}$ | 0.810 | 0.984 | 0.988 | **0.844** | 0.979 | 0.980 | 0.684 | 0.981 | 0.906 |
| HFI$_{LPIPS}$ (ours) | 0.796 | 0.988 | 0.992 | 0.804 | 0.997 | 0.998 | 0.807 | 0.998 | 0.923 |
| HFI$_{LPIPS_2}$ (ours) | **0.896** | **0.996** | **0.994** | 0.818 | **0.999** | **0.999** | **0.870** | **0.999** | **0.946** |

Table 4: AI-generated image detection performance (AUROC/AUPR) of HFI and training-free baselines in the DiffusionFace (Chen et al., 2024c) dataset. **Bold** and underline denotes the best and second best methods.

| Method | SDv1.5 T2I | SDv2.1 T2I | SDv1.5 I2I | SDv2.1 I2I | Mean |
|---|---|---|---|---|---|
| RIGID | 0.697/0.620 | **0.606**/0.531 | 0.532/0.529 | 0.539/0.542 | 0.594/0.556 |
| AEROBLADE$_{LPIPS}$ | 0.852/0.832 | 0.515/0.486 | 0.920/0.910 | 0.631/**0.613** | 0.730/0.710 |
| AEROBLADE$_{LPIPS_2}$ | 0.845/0.813 | 0.474/0.459 | 0.971/0.964 | 0.590/0.579 | 0.720/0.704 |
| HFI$_{LPIPS}$ (ours) | 0.870/**0.878** | 0.543/**0.584** | 0.944/0.942 | **0.639**/0.599 | 0.749/**0.751** |
| HFI$_{LPIPS_2}$ (ours) | **0.876**/0.853 | 0.522/0.536 | **0.987**/**0.985** | 0.620/0.592 | **0.751**/0.742 |

In the second scenario, we also report the performance of RIGID (He et al., 2024), a recently proposed training-free detection method. We also report the performance of competitive training-based methods: NPR (Tan et al., 2024a) and DRCT (Chen et al., 2024a). Both methods are trained on ImageNet data and SDv1.4-generated data.

**Design choices and implementation details** All experiments on HFI are done in fixed choices of the distance function $d$, the low-pass filter $\mathcal{F}$, and the ensemble strategy. For the distance function $d$, we use the standard LPIPS distance and that of AEROBLADE and denote the performance as HFDD$_{LPIPS}$ and HFDD$_{LPIPS_2}$, respectively. For the low pass-filter $\mathcal{F}$, We use a $3 \times 3$ Gaussian filter with a standard deviation of 0.8 consistent with Section 3.

We resize all images to fit the default dimension of the autoencoder (*i.e.*, $256 \times 256$ in MiniSD (Labs, 2022), and $512 \times 512$ for the rest). We implement the code on the PyTorch (Paszke et al., 2019) framework. We implement the training-based detection methods by testing the weight released by the authors on the official codebase. We refer to the Appendix for further details.

## 4.2 MAIN RESULTS

Table 1 and Table 2 report the performance of HFI and AEROBLADE in the cross-autoencoder setup on the GenImage and DiffusionFace dataset, respectively. We refer to Table 9 and Table 10 in the Appendix for the full results. HFI outperforms AEROBLADE on all benchmarks and autoencoder choices on average. It is noteworthy that HFI outperforms AEROBLADE in 61 out of 64 experiments and 26 out of 32 on the GenImage and DiffusionFace benchmarks, respectively.

Table 3 and Table 4 report the ensemble performance of HFI compared to various AI-generated image detection methods. HFI shows the best performance in most benchmarks and outperforms other training-free detection methods by a wide margin. Furthermore, HFI outperforms NPR (Tan et al., 2024a) and DRCT (Chen et al., 2024a) in 7 and 6 out of 8 experiments, respectively. Especially, HFI achieves near-perfect detection performance in SDv1.4, SDv1.5, and Wukong benchmarks where the underlying autoencoder is available.

We further analyze the failure cases of AEROBLADE and HFI where the underlying autoencoder of the generated data is the same as the autoencoder used in detection. Figure 3a and 3b report the 10 real samples and 10 SDv1.4-generated samples where AEROBLADE outputs the lowest and highest

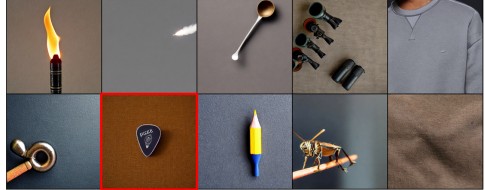

(a) Real images with low reconstruction error     (b) Generated images with high reconstruction error

Figure 3: **Visualization of the edge-cases. (a)** Visualization of the ImageNet data where AER-OBLADE outputs the smallest uncertainty. **(b)** Visualization of the SDv1.4-generated data where AEROBLADE outputs the highest uncertainty. We mark the sample where HFI also fails.

Table 5: Mean AI-generated image detection performance (AUPR) of HFI to the kernel size $k$ and standard deviation $\sigma$ of the Gaussian filter $\mathcal{F}$. We report the average performance computed across the GenImage dataset. **Bold** denotes the best hyperparameter.

| | AE: SDv2-base | | | AE: Kandinsky | | | AE: MiniSD | | |
|---|---|---|---|---|---|---|---|---|---|
| | $k=3$ | $k=5$ | $k=7$ | $k=3$ | $k=5$ | $k=7$ | $k=3$ | $k=5$ | $k=7$ |
| $\sigma = 0.5$ | 0.844 | 0.844 | 0.844 | 0.894 | 0.895 | 0.895 | 0.672 | 0.670 | 0.670 |
| $\sigma = 0.8$ | 0.898 | 0.897 | 0.897 | **0.900** | 0.895 | 0.895 | 0.700 | 0.695 | 0.695 |
| $\sigma = 1.1$ | **0.903** | 0.898 | 0.896 | 0.898 | 0.884 | 0.881 | 0.706 | 0.697 | 0.695 |
| $\sigma = 1.4$ | **0.903** | 0.897 | 0.892 | 0.895 | 0.875 | 0.866 | **0.707** | 0.695 | 0.689 |

uncertainty with the average of 0.003 and 0.029, respectively. Since the monotonous background observed in samples from Figure 3a is relatively easy to reconstruct, LPIPS distance is low in such samples. While HFI also shows low uncertainty in these samples, HFI successfully distinguishes most of the cases by lowering the average score of the generated data in Figure 3b *i.e.*, $0.029 \rightarrow -0.011$.

### 4.3 ABLATION STUDIES

**Design choices.** We extensively explore design choices for HFI under the GenImage dataset. First, we perform a hyperparameter analysis on the size of the kernel $k$ and the standard deviation $\sigma$ for the Gaussian blur filter $\mathcal{F}$ under different autoencoders. We report the result in Table 5. Note that $k = 3, \sigma = 0.8$ refers to the original setting. $k = 3$ performs the best consistently.

We further explore different choices of low-pass filtering. We apply box, bilateral, and median blur filters of kernel size 3 for the experiment. We also experiment with a discrete cosine transform (DCT) based frequency nulling scheme that erases high-frequency components above a given frequency. Since the size of the hyperparameter search space on the DCT is massive, we instead rely on our rule of thumb based on antialiasing. Namely, we view the encoder as a downsampling kernel and propose to set the cutoff frequency as the Nyquist frequency. We follow the practice of Chen et al. (2024b) for calculating the Nyquist frequency.

We report the result in Table 6. Surprisingly, the box filter also shows competitive performance, slightly outperforming the Gaussian filter on average. On the other hand, our designed hyperparameter on DCT underperforms over AEROBLADE. We show further experiment results in the Appendix.

We also experiment with different choices of perceptual distance. For the alternative distance, we experiment $LPIPS_1, LPIPS_3, LPIPS_4, LPIPS_5$, and DISTS (Ding et al., 2020). We report the ensemble performance consistent with the results of Table 3. We report the results in Table 7. $LPIPS_2$ shows the best performance.

**Robustness to corruption.** While our HFI has achieved great success on various benchmarks, the underlying high-frequency information can be affected by corruption. For example, applying JPEG (Wallace, 1992) transform to the image generates high-frequency artifacts to the corrupted image. To better understand our limitations on severe corruptions, we test HFI when the AI-generated and real data are corrupted. We follow the experiment setup of Frank et al. (2020) where JPEG compression, crop, Gaussian blur, and Gaussian noise are applied for corruption. We test HFI and AEROBLADE on the ImageNet vs SDv1.4 task with the autoencoder of SDv1.4 as the AE.

Table 6: Ablation studies of HFI under different low-pass filter. We report the ensemble performance. **Bold** denotes the best mapping choice.

| $\mathcal{F}$ | ADM | BigGAN | GLIDE | Midj | SD1.4 | SD1.5 | VQDM | Wukong | Mean |
|---|---|---|---|---|---|---|---|---|---|
| Gaussian blur | 0.796 | **0.988** | **0.992** | **0.804** | 0.997 | 0.998 | 0.807 | **0.998** | 0.923 |
| Box blur | 0.819 | 0.986 | **0.992** | 0.801 | **0.998** | **0.999** | 0.805 | **0.998** | **0.925** |
| Bilateral blur | 0.738 | 0.912 | 0.920 | 0.700 | 0.970 | 0.951 | 0.635 | 0.974 | 0.850 |
| Median blur | **0.823** | 0.975 | 0.973 | 0.797 | 0.997 | 0.996 | **0.808** | 0.997 | 0.921 |
| DCT | 0.661 | 0.842 | 0.954 | 0.768 | 0.941 | 0.940 | 0.579 | 0.958 | 0.830 |

Table 7: Ablation studies of HFI under different distance functions. We report the ensemble performance. **Bold** denotes the best distance choice.

| $d$ | ADM | BigGAN | GLIDE | Midj | SD1.4 | SD1.5 | VQDM | Wukong | Mean |
|---|---|---|---|---|---|---|---|---|---|
| LPIPS | 0.796 | 0.988 | 0.992 | 0.804 | 0.997 | 0.998 | 0.807 | 0.998 | 0.923 |
| $LPIPS_1$ | **0.903** | 0.994 | **0.995** | 0.779 | 0.986 | 0.987 | **0.893** | 0.987 | 0.941 |
| $LPIPS_2$ | 0.896 | **0.996** | 0.994 | **0.818** | **0.999** | **0.999** | 0.870 | **0.999** | **0.946** |
| $LPIPS_3$ | 0.854 | 0.989 | 0.992 | 0.768 | 0.987 | 0.988 | 0.805 | 0.996 | 0.922 |
| $LPIPS_4$ | 0.672 | 0.602 | 0.813 | 0.622 | 0.909 | 0.910 | 0.632 | 0.910 | 0.759 |
| $LPIPS_5$ | 0.586 | 0.539 | 0.686 | 0.607 | 0.894 | 0.798 | 0.555 | 0.780 | 0.681 |
| DISTS | 0.638 | 0.749 | 0.814 | 0.606 | 0.746 | 0.742 | 0.551 | 0.740 | 0.698 |

We report the result in Figure 4. The trends can be divided into two cases. When the corruption is image-dependent, our method is relatively robust in small corruptions with larger gaps against AEROBLADE than when evaluated in clean images (*e.g.*, Crop (Figure 4b), Blur (Figure 4c)). However, our method struggles in the case of input-independent Gaussian Noise (Figure 4d) where the high-frequency distortion is applied to both images equally.

Since low-pass filtering has been established as a standard practice for mitigating the effects of external noise, we also experiment with $\text{B-HFI}(\mathbf{x}) = \text{HFI}(\mathcal{F}_B(\mathbf{x}))$. We use a Gaussian filter with $k = 3$ and $\sigma = 0.8$ for $\mathcal{F}_B$. We also report their result in Figure 4 denoted as B-HFI. The B-HFI method enhances robustness compared to HFI while outperforming AEROBLADE on the clean data.

## 5 TRACING LDM-GENERATED IMAGES

We further elaborate on tracing LDM-generated images introduced in Section 2.3. To be specific, for a given LDM model $\mathcal{M}_1$ and its autoencoder $\text{AE}_{\mathcal{M}_1}$, we directly apply $\text{HFDD}_{\text{AE}_{\mathcal{M}_1}}$ to distinguish the belonging images generated from $\mathcal{M}_1$ from the others. Namely, we regard the input image $\mathbf{x}$ as generated from the model $\mathcal{M}_1$ if $\text{HFDD}_{\text{AE}_{\mathcal{M}_1}}$ outputs low score.

We follow the practice of Wang et al. (2024) and consider the case where both $\mathcal{M}_1$ and $\mathcal{M}_2$ are from 4 different LDMs: SDv1.5, SDv2-base, SDv2.1, and Kandinsky2.1. For experiments, we follow the setup of Wang et al. (2024) where they sample 54 prompts and generate 10 images per prompt, resulting in 540 images per model. We also compare against LatentTracer (Wang et al., 2024), the best baseline that performs input optimization in the test time. We test LatentTracer on the official code released by the authors. We report the performance of $\text{HFDD}_{\text{LPIPS}_2}$ and $\text{AEROBLADE}_{\text{LPIPS}_2}$.

We report the result in Table 8. Both HFI and LatentTracer achieve near-perfect detection performance. On the other hand, AEROBLADE struggles in "SD vs Kandinsky" tasks. It is worth noting that HFI is much more efficient in time complexity than LatentTracer. Namely, HFI takes 0.255s/sample, which achieves 57x speedup against the LatentTracer which takes 14.65s/sample in A100 gpus.

## 6 RELATED WORKS

**Training-based AI-generated image detection.** Most training-based AI-generated image detection methods follow a two-stage setup: the key representation is extracted from the image followed by training a binary classifier over the key representations. A variety of representations have been

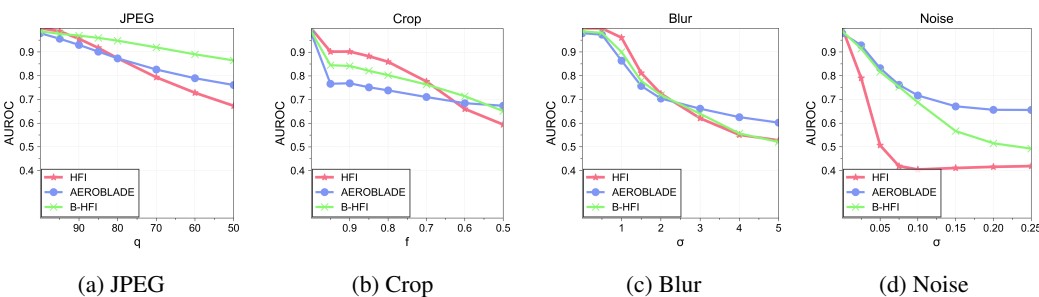

|  |  |  |  |
|---|---|---|---|
| (a) JPEG | (b) Crop | (c) Blur | (d) Noise |

Figure 4: Performance of HFI, AEROBLADE, and B-HFI under corruption.

Table 8: Belonging vs non-belonging image detection performance (AUPR) of HFI compared to LatentTracer (Wang et al., 2024). We denote $\mathcal{M}_1$ and $\mathcal{M}_2$ to the belonging and non-belonging model, respectively. **Bold** denotes the best method.

| | $\mathcal{M}_1$: SDv1.5 | | | $\mathcal{M}_1$: SDv2-base | | | $\mathcal{M}_1$: Kandinsky | | |
|---|---|---|---|---|---|---|---|---|---|
| Method | SDv2-base | SDv2.1 | Kand | SDv1.5 | SDv2.1 | Kand | SDv1.5 | SDv2-base | SDv2.1 |
| LatentTracer | 0.9990 | 0.9983 | **0.9976** | **0.9994** | 0.9990 | 0.9984 | 0.9973 | 0.9971 | 0.9945 |
| AEROBLADE$_{\text{LPIPS}_2}$ | 0.9652 | 0.9935 | 0.8145 | 0.9392 | 0.9941 | 0.8271 | 0.9950 | 0.9982 | 0.9987 |
| HFI$_{\text{LPIPS}_2}$ (ours) | **0.9999** | **1.0000** | 0.9972 | 0.9985 | **0.9998** | **0.9989** | **0.9986** | **0.9986** | **0.9988** |

proposed, including handcrafted spatial features (Popescu & Farid, 2005), patch statistics (Chai et al., 2020), a feature of the foundation model (Ojha et al., 2023), or their unification (Yan et al., 2024).

Leveraging frequency maps as a key representation has also been a major practice of training-based AI-generated image detection methods. Frank et al. (2020) discover that the GAN-generated images contain noticeable frequency artifacts. Li et al. (2021) fuse the spatial and frequency representation and adopt a single-center loss for pulling out manipulated face representation. Wang et al. (2023a) introduce a relation module that connects multiple domains. Tan et al. (2024b) incorporate the FFT module inside the classifier. Although these methods operate in the frequency domain, our findings indicate that spatial low-pass filtering may provide a more effective mapping.

**Training-free AI-generated image detection.** Since no training data is available, training-free AI-generated image detection methods utilize a pre-trained model to design the uncertainty function. Ma et al. (2023) utilize the error of deterministic forward-reverse diffusion process to detect diffusion-generated images. Ricker et al. (2024) utilize the perceptual distance between the input and its reconstruction through the autoencoder of LDMs to detect LDM-generated images. He et al. (2024) utilizes the cosine similarity on the feature of Dinov2 (Oquab et al., 2024) between the input data and its perturbation. The perturbation is constructed by injecting Gaussian Noise into the input.

**Aliasing in image domain.** Zhang (2019) discusses aliasing in the classification task that aliasing occurs while in subsampling and design a blurring-based pooling operation for the remedy. Vasconcelos et al. (2021) propose the variant by positioning the blurring operation before the non-linearities. Karras et al. (2021) propose a custom upsampling layer to deal with aliasing while training generative adversarial networks. Agnihotri et al. (2024) also modify the operation in the upsampling layer in the pixel-wise prediction task. However, to our knowledge, most of the existing works focus on the training phase.

# 7 CONCLUSION

This paper proposes HFI, a novel, training-free AI-generated image detection method based on the autoencoder of LDM. HFI measures the influence of the high-frequency spatial component on the distortion of the reconstructed image through the autoencoder. HFI can effectively distinguish 3 types of distributions: images generated from the given autoencoder, other AI-generated images, and real images. Hence, HFI achieves the state-of-the-art in both training-free AI-generated image detection and generated image detection of the specified model.

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

Table 9: Cross-autoencoder AI-generated image detection performance (AUROC/AUPR) of HFI and AEROBLADE (Ricker et al., 2024) in the GenImage (Zhu et al., 2023) dataset. **Bold** denotes the best method.

| Method | ADM | BigGAN | GLIDE | Midj | SD1.4 | SD1.5 | VQDM | Wukong | Mean |
|---|---|---|---|---|---|---|---|---|---|
| *AE: SDv1.4 (Rombach et al., 2022)* | | | | | | | | | |
| AEROBLADE$_{\text{LPIPS}}$ | 0.803/0.755 | 0.885/0.907 | 0.974/0.975 | 0.849/0.847 | 0.979/0.985 | 0.980/0.986 | 0.637/0.593 | 0.982/0.988 | 0.886/0.880 |
| AEROBLADE$_{\text{LPIPS}_2}$ | 0.855/0.831 | 0.980/0.986 | 0.989/0.989 | 0.858/0.860 | 0.982/0.987 | 0.983/0.988 | 0.730/0.711 | 0.983/0.989 | 0.920/0.918 |
| HFI$_{\text{LPIPS}}$ (ours) | 0.883/0.852 | 0.992/0.995 | **0.995/0.995** | **0.874/0.866** | **1.0/1.0** | **0.999/1.0** | 0.788/0.763 | **0.999/0.999** | 0.941/0.934 |
| HFI$_{\text{LPIPS}_2}$ (ours) | **0.916/0.921** | **0.994/0.997** | 0.994/**0.995** | 0.852/0.841 | 0.999/0.999 | 0.999/0.999 | **0.858/0.885** | **0.999/0.999** | **0.951/0.955** |
| *AE: SDv2-base (Rombach et al., 2022)* | | | | | | | | | |
| AEROBLADE$_{\text{LPIPS}}$ | 0.799/0.752 | 0.903/0.920 | 0.976/0.976 | 0.849/0.849 | 0.817/0.846 | 0.821/0.849 | 0.626/0.584 | 0.837/0.863 | 0.829/0.830 |
| AEROBLADE$_{\text{LPIPS}_2}$ | 0.855/0.833 | 0.978/0.986 | 0.988/0.989 | 0.860/**0.863** | 0.806/0.841 | 0.809/0.841 | 0.724/0.709 | 0.832/0.864 | 0.857/0.866 |
| HFI$_{\text{LPIPS}}$ (ours) | 0.881/0.857 | 0.987/0.991 | 0.993/0.994 | **0.865/0.860** | **0.883/0.902** | **0.890/0.905** | 0.807/0.774 | 0.887/0.906 | 0.899/0.899 |
| HFI$_{\text{LPIPS}_2}$ (ours) | **0.920/0.928** | **0.993/0.996** | **0.995/0.996** | 0.856/0.849 | 0.859/0.882 | 0.865/0.884 | **0.859/0.887** | **0.890/0.914** | **0.905/0.917** |
| *AE: Kandinsky (Razzhigaev et al., 2023)* | | | | | | | | | |
| AEROBLADE$_{\text{LPIPS}}$ | 0.799/0.754 | 0.876/0.889 | 0.975/0.976 | 0.856/0.855 | 0.683/0.712 | 0.686/0.714 | 0.640/0.597 | 0.712/0.733 | 0.777/0.779 |
| AEROBLADE$_{\text{LPIPS}_2}$ | 0.864/0.846 | 0.980/0.986 | 0.989/0.990 | 0.867/0.870 | 0.661/0.692 | 0.662/0.691 | 0.746/0.734 | 0.702/0.731 | 0.809/0.818 |
| HFI$_{\text{LPIPS}}$ (ours) | 0.896/0.875 | 0.991/0.994 | **0.996/0.997** | **0.898/0.904** | **0.854/0.879** | **0.858/0.881** | 0.819/0.802 | **0.848/0.868** | **0.895/0.900** |
| HFI$_{\text{LPIPS}_2}$ (ours) | **0.937/0.947** | **0.993/0.996** | 0.994/0.996 | 0.872/0.872 | 0.763/0.792 | 0.767/0.792 | **0.892/0.920** | 0.803/0.830 | 0.878/0.893 |
| *AE: MiniSD (Labs, 2022)* | | | | | | | | | |
| AEROBLADE$_{\text{LPIPS}}$ | 0.658/0.623 | 0.598/0.586 | 0.887/0.866 | 0.766/0.755 | 0.581/0.592 | 0.586/0.601 | 0.587/0.567 | 0.613/0.610 | 0.660/0.650 |
| AEROBLADE$_{\text{LPIPS}_2}$ | 0.672/0.620 | 0.722/0.748 | 0.929/0.922 | **0.792/0.786** | 0.543/0.559 | 0.545/0.565 | 0.531/0.497 | 0.600/0.604 | 0.667/0.663 |
| HFI$_{\text{LPIPS}}$ (ours) | **0.679/0.631** | 0.695/0.716 | 0.949/0.939 | 0.736/0.696 | 0.646/**0.630** | 0.644/**0.633** | **0.684**/0.673 | 0.700/0.683 | 0.717/0.700 |
| HFI$_{\text{LPIPS}_2}$ (ours) | 0.665/0.590 | **0.887/0.919** | **0.977/0.966** | 0.789/0.756 | **0.647/0.630** | **0.648**/0.627 | 0.672/**0.677** | **0.727/0.722** | **0.752/0.736** |

Table 10: Cross-autoencoder AI-generated image detection performance (AUROC/AUPR) of HFI and AEROBLADE (Ricker et al., 2024) in the DiffusionFace (Chen et al., 2024c) dataset. **Bold** denotes the best method.

| Method | SDv1.5 T2I | SDv2.1 T2I | SDv1.5 I2I | SDv2.1 I2I | Mean |
|---|---|---|---|---|---|
| *AE: SDv1.4 (Rombach et al., 2022)* | | | | | |
| AEROBLADE$_{\text{LPIPS}}$ | 0.884/0.860 | 0.507/0.477 | 0.978/0.972 | **0.630/0.612** | 0.750/0.730 |
| AEROBLADE$_{\text{LPIPS}_2}$ | 0.865/0.833 | 0.464/0.452 | 0.986/0.982 | 0.600/0.586 | 0.729/0.713 |
| HFI$_{\text{LPIPS}}$ (ours) | **0.922/0.925** | **0.560/0.583** | 0.980/0.979 | 0.622/0.591 | **0.772/0.770** |
| HFI$_{\text{LPIPS}_2}$ (ours) | 0.899/0.873 | 0.513/0.523 | **0.993/0.992** | 0.605/0.582 | 0.753/0.743 |
| *AE: SDv2-base (Rombach et al., 2022)* | | | | | |
| AEROBLADE$_{\text{LPIPS}}$ | 0.852/0.831 | 0.514/0.486 | 0.837/0.821 | 0.631/**0.613** | 0.709/0.688 |
| AEROBLADE$_{\text{LPIPS}_2}$ | 0.848/0.813 | 0.487/0.467 | 0.875/0.861 | 0.609/0.595 | 0.705/0.684 |
| HFI$_{\text{LPIPS}}$ (ours) | 0.860/**0.865** | **0.593/0.578** | 0.888/0.873 | **0.640**/0.600 | 0.732/**0.729** |
| HFI$_{\text{LPIPS}_2}$ (ours) | **0.871**/0.843 | 0.518/0.532 | **0.933/0.925** | 0.622/0.595 | **0.736**/0.724 |
| *AE: Kandinsky (Razzhigaev et al., 2023)* | | | | | |
| AEROBLADE$_{\text{LPIPS}}$ | 0.846/0.827 | 0.509/0.484 | 0.637/0.619 | 0.611/**0.595** | 0.651/0.631 |
| AEROBLADE$_{\text{LPIPS}_2}$ | 0.845/0.812 | 0.466/0.453 | 0.692/0.666 | 0.589/0.578 | 0.648/0.627 |
| HFI$_{\text{LPIPS}}$ (ours) | **0.925/0.925** | **0.601/0.644** | 0.760/0.735 | **0.623**/0.586 | **0.727/0.723** |
| HFI$_{\text{LPIPS}_2}$ (ours) | 0.882/0.858 | 0.505/0.514 | **0.779/0.758** | 0.594/0.575 | 0.690/0.676 |
| *AE: MiniSD (Labs, 2022)* | | | | | |
| AEROBLADE$_{\text{LPIPS}}$ | 0.838/0.803 | 0.517/0.478 | 0.439/0.456 | 0.610/0.594 | 0.601/0.583 |
| AEROBLADE$_{\text{LPIPS}_2}$ | 0.844/0.808 | 0.506/0.467 | 0.489/0.492 | **0.615/0.601** | 0.614/0.592 |
| HFI$_{\text{LPIPS}}$ (ours) | **0.918/0.915** | **0.667/0.646** | 0.587/0.571 | 0.599/0.577 | **0.693**/0.677 |
| HFI$_{\text{LPIPS}_2}$ (ours) | 0.917/0.907 | 0.605/0.577 | **0.638/0.637** | 0.609/0.589 | 0.692/**0.678** |

## A    FULL RESULTS ON THE CROSS-AUTOENCODER SETUP

We refer to Table 9 and 10 for the full results of HFI compared to AEROBLADE.

Table 11: Ablation studies of HFI with discrete cosine transform as F under different hyperparameters on cutoff frequency $f$. We report the ensemble performance. **Bold** denotes the best hyperparameter choice.

| $f$ | ADM | BigGAN | GLIDE | Midj | SD1.4 | SD1.5 | VQDM | Wukong | Mean |
|---|---|---|---|---|---|---|---|---|---|
| 8 | **0.716** | **0.879** | **0.966** | **0.824** | **0.956** | **0.957** | 0.578 | **0.963** | **0.855** |
| 16 | 0.686 | 0.848 | 0.952 | 0.799 | 0.937 | 0.938 | 0.562 | 0.948 | 0.834 |
| 24 | 0.686 | 0.860 | 0.956 | 0.793 | 0.937 | 0.937 | 0.571 | 0.951 | 0.836 |
| 32 | 0.676 | 0.854 | 0.956 | 0.778 | 0.939 | 0.939 | 0.574 | 0.956 | 0.834 |
| 36 | 0.661 | 0.842 | 0.954 | 0.768 | 0.941 | 0.940 | **0.579** | 0.958 | 0.830 |
| 40 | 0.644 | 0.825 | 0.951 | 0.756 | 0.939 | 0.940 | 0.571 | 0.959 | 0.823 |
| 44 | 0.625 | 0.805 | 0.945 | 0.742 | 0.933 | 0.933 | 0.562 | 0.955 | 0.813 |
| 48 | 0.606 | 0.770 | 0.936 | 0.727 | 0.920 | 0.920 | 0.553 | 0.948 | 0.798 |

# B  FURTHER DETAILS ON THE EXPERIMENT

**Datasets.** For the further information, diffusionface's T2I and I2I datasets constitutes of 90000 data each. For fair comparison against the real dataset with the 30000 dataset, we take a subset of the diffusionface with the 30000 dataset. We provide each link for the specified dataset we used in the footnote for SDv1.5 T2I [1], SDv2.1 T2I[2], SDv1.5 I2I [3], and SDv2.1 I2I[4] dataset all downloadable through the author's official GitHub repository [5]. The whole GenImage dataset is downloaded through the author's GitHub repository [6].

**Baselines.** First, we elaborate the specified link we used to evaluate the training-based AI-generated image detection methods in the footnote for DRCT (Chen et al., 2024a) [7] and NPR (Tan et al., 2024a) [8]. Note that we directly use the author's checkpoint following the data augmentation strategies noted in the corresponding paper. Furthermore, for evaluating the baseline performance of LatentTracer (Wang et al., 2024), we use the author's official code [9].

# C  FURTHER HYPERPARAMETER SEARCH RESULTS ON THE DISCRETE COSINE TRANSFORM

We further explore the available choices of the hyperparameter $f$ which corresponds to the cutoff frequency of erasing high-frequency components in the frequency map computed by the discrete cosine transform. Note that $f = 36$ corresponds to the Nyquist frequency $f = 64/\sqrt{3} \approx 36.95$.

---

[1] `stable_diffusion_v_1_5_text2img_p3g7.tar`
[2] `stable_diffusion_v_2_1_text2img_p0g5.tar`
[3] `SDv15_DS0.3.tar`
[4] `SDv21_DS0.3.tar`
[5] `https://github.com/Rapisurazurite/DiffFace`
[6] `https://github.com/GenImage-Dataset/GenImage`
[7] `https://github.com/beibuwandeluori/DRCT`
[8] `https://github.com/chuangchuangtan/NPR-DeepfakeDetection`
[9] `https://github.com/ZhentingWang/LatentTracer`

