# OpenReview forum: "Training-free Detection of AI-generated Images via High-frequency Influence"
_ICLR.cc/2025/Conference — ICLR 2025 Conference Withdrawn Submission_

### Official Review · Reviewer_gwxt · 2024-10-23

**Soundness:** 3
**Presentation:** 3
**Contribution:** 2
**Rating:** 5
**Confidence:** 4

**Summary:**

This paper proposes a method for distinguishing AI-generated images from real ones. The authors suggest that the level of aliasing detected in reconstructed images produced by the autoencoder of LDM can serve as a criterion for determining whether an image is AI-generated, and the effectiveness of the proposed method is validated through a series of experiments.

**Strengths:**

1. Using the level of aliasing detected in reconstructed images produced by the autoencoder to achieve training-free AI-generated content detection shows good insight.
2. The experimental section is thorough, with detailed ablation studies and comprehensive comparisons across different benchmark models.

**Weaknesses:**

1. The technical contribution is limited. The proposed method is quite simple and essentially relies on the tokenizer's ability to achieve the expected recognition effect. How would improvements in tokenizer reconstruction quality affect the performance of HFI? Is there a point at which HFI would no longer be effective?
2. The motivation and solution idea is very similar to AEROBLADE (CVPR 24) in solving the problem, and the novelty is insufficient because it is just a simple strategy to find out the artifacts of the VAE. It feels more like a technical report than a new model.
3. Current generative methods are more diverse, including autoregressive and mask-based methods, not just LDMs. Have the authors considered how HFI might be applied to or modified for autoregressive or mask-based generative methods?

**Questions:**

See the weaknesses above.

---

### Official Review · Reviewer_Mdhe · 2024-10-23

**Soundness:** 3
**Presentation:** 3
**Contribution:** 3
**Rating:** 6
**Confidence:** 3

**Summary:**

The paper presents a training-free score function for separating between real images and images generated by deep generative models. The core is based on the observation that the autoencoders (AEs) used for latent diffusion models (LDMs) have limited capabilities in encoding and decoding high-frequency information. The score function is formulated as the difference in reconstruction error of the original and a low-pass filtered test image. A high score means that there is a large discrepancy in error between the original and filtered image, indicating that high-frequency information is lost through the AE. Experiments compare the method against previous training-free and data-driven detection methods on two datasets, GenImage and DiffusionFace. Results are presented for using different AEs as well as ensembles of AEs, and investigating differences between different generative models. An ablation study compares different settings in terms of formulation of low-pass filtering and the distance measure used. In addition, a study is presented for analyzing the performance under different types of image degradations.

**Strengths:**

The topic is highly relevant today, and the results indicate favorable performance over previous work. The method itself is simple and straightforward, which makes it a good candidate for general purpose detection of AI-generated image content. The specific formulation is also original as far as I know. The paper is well-written and easy to follow. It also presents a a multitude of interesting experiments, facilitating assessing the performance from different perspectives. The paper is of relatively high significance in the sense that it presents a good combination of simplicity, favorable results, and highly relevant topic.

**Weaknesses:**

The core technical contribution is relatively small, where previous work also have explored the AE reconstruction error for detection of AI-generated images. Although the simplicity itself is a good thing, it is also based on an empirical observation. That is, the core methodology is not really backed up from analyzing the capabilities of the encoder/decoder in terms of compressing/generating high-frequency information. It would be interesting with either a deeper empirical study to verify this behavior, or by providing some theoretical bound of the spectra that can be encoded and decoded.

Another weakness is the sensitivity to image artifacts (although this is likely a problem for most previous methods as well), especially compression artifacts that are common in real-world scenarios. Also, it seems that the detection problem is increasingly more difficult with recent models (for example, Midjourney in Table 3 and SDv2.1 in Table 4). That is, will the presented method be a viable approach in the future or will it quickly become obsolete? A broader discussion in these directions would be valuable.

Although there are many valuable experiments, from different perspectives, some dimensions that would be interesting to explore more explicitly are:
- Results when there is a discrepancy between the data distribution of test images compared to the images used to train the AE.
- Comparisons for different categories of test images, e.g., in terms of their frequency distribution.

**Questions:**

* A real image encoded and decoded by the AE would also get a low score? That is, the measure specifically targets the capabilities of the AE, not the generative process itself. How does this relate to the seemingly good generalization to models not relying on an explicit decoder?
* How difficult would it be to enhance or adapt images to "fool" the score function, e.g., increasing the amount of high-frequency information by means of super-resolution or similar?
* On the same lines; would it be possible to use HFI as a training objective to promote generation of images that are more difficult to detect as AI-generated? Or is there an inherent limitation in the AE that will prevent this?
* Can the score be measured without the AE, i.e. by directly analyzing the amount of high-frequency information in the test image?
* There is a certain focus on LDMs (abstract, etc.), but in the end the method seems applicable to other generative models, such as BigGAN, GLIDE, etc. Perhaps this should be rephrased?
* It was interesting to see examples with edge-cases. However, these were for AEROBLADE. How about edge-cases for HFI?
* The acronym HFDD is used in some explanations. What is this referring to? Or is it supposed to be HFI?

---

### Official Review · Reviewer_2u5z · 2024-10-26

**Soundness:** 3
**Presentation:** 3
**Contribution:** 3
**Rating:** 6
**Confidence:** 3

**Summary:**

This paper proposes a training-free method for AI-generated image detection. It is motivated by the observation that traditional downsampling-upsampling architectures leave obvious high-frequency flaws so that we can distinguish real/fake images by observing the occurrence of the flaws after being reconstructed.

**Strengths:**

1. The writing is good. The authors present their motivation and the corresponding evidence reasonably. And the description of the method is professional.
2. The method is effective and efficient. It needs no training data and avoids performance bias caused by collecting training data. It achieves significant performance improvement in GenImage this challenging benchmark.
3. The various ablation studies are conducted in the paper.

**Weaknesses:**

1. In the experiments, some training-free methods are missed, such as DIRE, RECCE.
2. Considering that the method is based on high-frequency flaws, I think it cannot achieve good robustness against denoising attacks.
3. The metrics are AUROC/AUPR, and the authors reported the results for each generator individually. I think different generators correspond to different classification thresholds. If all images in GenImage are placed together, can the method also classify them successfully? The authors didn't clarify how to determine a threshold for classification in the paper.  I think it is an important problem for this method, especially when considering applying it in practical applications.

**Questions:**

1. I didn't understand the reason for using Eq.3. The authors mention that "both components of the distance function are dependent on the input". Could the authors interpret it in detail?
2. The meaning of < . , . > is not given. Does it indicate the inner-product?

---

### Official Review · Reviewer_mJ7p · 2024-11-03

**Soundness:** 2
**Presentation:** 3
**Contribution:** 2
**Rating:** 5
**Confidence:** 3

**Summary:**

This paper introduces a training-free detection score function, termed High-Frequency Influence (HFI), for distinguishing between real and AI-generated images. The proposed method is validated with extensive experiments on the GenImage and DiffusionFace benchmarks.

**Strengths:**

1. The observation that autoencoders often misrepresent high-frequency details in real data is intriguing.
2. The paper presents extensive experiments, demonstrating that the proposed method outperforms existing baselines.
3. The method is straightforward, making it easy to reproduce.

**Weaknesses:**

1. The proposed method is based on the observation of distortions in high-frequency details. However, the authors use only 162 ImageNet images from the “Jaguar” class to verify this observation and hypothesis. Validation on a larger dataset, along with statistical analysis, is necessary, as the current sample size may not sufficiently support the hypothesis.
2. Regarding the robustness of the algorithm: Recent works have noticed the loss of high-frequency information during compression and proposed solutions, I am curious about whether HFI would remain effective when such solutions are applied, such as the method in FA-VAE[1].
3. There are issues with clarity and rigor in defining and presenting the formulas in the paper.

[1] Lin, X., Li, Y., Hsiao, J., Ho, C. and Kong, Y., 2023. Catch Missing Details: Image reconstruction with frequency augmented variational autoencoder. In Proceedings of the IEEE/CVF Conference on Computer Vision and Pattern Recognition (pp. 1736-1745).

**Questions:**

1. Concerning Eq. (1): The motivation and definition of Eq. (1) are unclear and may raise readers' questions. Why do the authors compute the inner product between the gradient term and the difference term? The first term represents the rate of change of the distance between the input and its autoencoder reconstruction, while the second term is the difference between the input and a low-pass-filtered version. What is the significance of this inner product? Furthermore, the definition of the reconstruction distance function lacks clarity, and it is unclear whether $\nu$ represents the distribution of the input data or a specific dataset. Also, since $F(x)$ is a low-pass filter, why not directly use a high-pass filter to extract high-frequency information?
2. Concerning Eq. (2): The term $\arg\min_{\mathcal{D}, \mathcal{E}}$ in the equation refers to optimization over the encoder $\mathcal{E}$ and decoder $\mathcal{D}$. Both $\mathcal{D}$ and $\mathcal{E}$ have parameter sets (such as $\theta_{\mathcal{E}}$ and $\theta_{\mathcal{D}}$). A more formal notation would be $\arg\min_{\theta_{\mathcal{E}}, \theta_{\mathcal{D}}}$.
3. Regarding Eq. (3): How is the threshold $\tau$ chosen?
4. On generalizability: Would this algorithm still be effective for GAN-based models?
5. Why does B-HFI achieve higher AUROC in certain cases of corruption, given that $F_B$ has already filtered out high-frequency information from the input? Could the authors provide further analysis on this?

---

### Note · Authors · 2024-11-15

I have read and agree with the venue's withdrawal policy on behalf of myself and my co-authors.